# The Vital Role of the *CAMTA* Gene Family in *Phoebe bournei* in Response to Drought, Heat, and Light Stress

**DOI:** 10.3390/ijms25189767

**Published:** 2024-09-10

**Authors:** Kehui Zheng, Min Li, Zhicheng Yang, Chenyue He, Zekai Wu, Zaikang Tong, Junhong Zhang, Yanzi Zhang, Shijiang Cao

**Affiliations:** 1College of Computer and Information Sciences, Fujian Agriculture and Forestry University, Fuzhou 350002, China; zhkehui@fafu.edu.cn (K.Z.); yue15260953616@163.com (C.H.); 2College of Life Sciences, Fujian Agriculture and Forestry University, Fuzhou 350002, China; lm13138488614@163.com; 3College of Future Technologiesm, Fujian Agriculture and Forestry University, Fuzhou 350002, China; 18359797810@163.com; 4College of Forestry, Fujian Agriculture and Forestry University, Fuzhou 350002, China; wujingyu183@gmail.com; 5State Key Laboratory of Subtropical Silviculture, School of Forestry & Biotechnology, Zhejiang A&F University, Lin’an, Hangzhou 311300, China; zktong@zafu.edu.cn (Z.T.); zhangjunhong@zafu.edu.cn (J.Z.); 6Center for Metabolomics, Haixia Institute of Science and Technology, Fujian Agriculture and Forestry University, Fuzhou 350002, China

**Keywords:** *Phoebe bournei*, *CAMTA*, gene family, abiotic stress, expression analysis

## Abstract

The calmodulin-binding transcriptional activator (CAMTA) is a small, conserved gene family in plants that plays a crucial role in regulating growth, development, and responses to various abiotic stress. Given the significance of the *CAMTA* gene family, various studies have been dedicated to uncovering its functional characteristics. In this study, genome-wide identification and bioinformatics analysis were conducted to explore *CAMTA*s in *Phoebe bournei*. A total of 17 *CAMTA* genes, each containing at least one domain from CG-1, TIG, ANK, or IQ, were identified in the *P. bournei* genome. The diversity of *PbCAMTAs* could be varied depending on their subcellular localization. An analysis of protein motifs, domains, and gene structure revealed that members within the same subgroup exhibited similar organization, supporting the results of the phylogenetic analysis. Gene duplications occurred among members of the *PbCAMTA* gene family. According to the *cis*-regulatory element prediction and protein–protein interaction network analysis, eight genes were subjected to qRT-PCR under drought, heat, and light stresses. The expression profiles indicated that *PbCAMTA*s, particularly *PbCAMTA2*, *PbCAMTA12*, and *PbCAMTA16*, were induced by abiotic stress. This study provides profound insights into the functions of *CAMTA*s in *P. bournei*.

## 1. Introduction

“Nanmu,” scientifically referred to as *Phoebe bournei*, is a distinctive and ornamental tree species that is native to China; it also plays a crucial role in soil enrichment and water conservation. Its importance as a national asset is underscored by its contribution to ecological sustainability [1,2]. *P. bournei* is renowned for its utility in crafting high-quality furniture and is widely regarded as an elite wood type for architectural construction, furniture production, artistic carving, and decorative applications. Because of increasingly severe environmental conditions, wild populations of *P. bournei* are extremely rare [3]. Therefore, it is particularly important to explore the effects of various abiotic factors that affect the growth and development of *P. bournei* and, thus, to restore *P. bournei* abundance in wild forests [4].

The divalent ion calcium (Ca^2+^) is an essential nutrient element relating to the growth and development of plants. As the most ubiquitous secondary messenger, it is involved in or triggers numerous cellular signaling pathways [5]. Ca^2+^ can exclusively bind with proteins with EF-hand domains, such as CaM (calmodulins, calmodulin-like proteins) [6,7], CBL (calcineurin-B-like proteins) [8], CDPK (Ca^2+^-dependent protein kinases), and CaMK (calcium and calcium/calmodulin-dependent protein kinase) [9]. Calmodulins represent the primary category of Ca^2+^ sensors, which are capable of modulating a wide array of cellular functions in plants, encompassing stress responses and developmental processes [10,11]. Ca^2+^ and the Ca^2+^/CaM complex can trigger multiple biological processes and are implicated in stress signaling through activating or inhibiting downstream transcription factors (TFs) [12,13]. While calmodulins are the most important type of Ca^2+^ receptor, the Ca^2+^/CaM complex can bind to enormous target proteins and, in so doing, regulate the development of plants.

The calmodulin-binding transcriptional activator (CAMTA) serves as a critical transcription factor that plays a vital role in regulating plant growth, development, and stress responses. These factors are strongly responsive to hormonal cues, including abscisic acid, salicylic acid, and auxin, as well as a wide range of environmental stressors such as arid conditions, salt stress, and temperature fluctuations [14,15,16]. CAMTAs exhibit structural conservation. These functional domain modules include: (1) the CG-1 DNA binding domain [17], (2) TIG (a transcription factor immunoglobulin), a domain involved in non-specific DNA binding [18], (3) the tandem repeat IQ motif (IQXXXRGXXX), a Ca^2+^-independent calmodulin-binding domain that can directly bind to CaM or CML (calmodulin-like) proteins [19,20], (4) the ankyrin repeats (ANK) domain, which can have immediate interactions between different proteins [7], and (5) CaMBD repeats that bind calmodulin in a calcium-dependent manner.

The *CAMTA* gene family has been extensively studied in various species [21]. Tobacco (*Nicotiana tabacum*) was the first of these species [22]. *CAMTA* was also systemically characterized in other plants, such as *Arabidopsis thaliana* [23], rice (*Oryza sativa*) [24], maize (*Zea mays*) [25], soybean (*Glycine max*) [26], and land cotton (*Gossypium hirsutum*) [27], which have all been thoroughly investigated. Recent research has also found *CAMTAs* in *Solanaceae (Solanum melongena*, *Solanum lycopersicum*, *Solanum pennellii*, *Capsicum annuum*, and *Lycium barbarum)* [28], *Heimia myrtifolia* [29], peach (*Prunus persica*) [30], and wheat (*Triticum aestivum*) [31]. It was found that *CAMTA* gene family members are triggered by many exogenous (heat, salinity, cold, drought) or endogenous (hormones) stresses [32,33,34]. *CAMTA1* in *A. thaliana* can address drought-related challenges by altering the expression patterns of genes associated with the response and generating ABA [35]. *GmCAMTA12* of *Glycine max* is an important transcription factor that can be exploited to manage drought stress by regulating downstream genes [36]. *AtCAMTA6* is involved in modulating the transcription of genes in germinating seeds, which are key to maintaining the Na^+^ balance and enduring salt stress resistance. The expression of all *CAMTA* genes in tea plants was upregulated after 72 h of NaCl treatment [37]. Under heat conditions, the *GhCAMTA11* gene from *Gossypium hirsutum* exhibits exclusive expression within the root tissues [27]. Most *CAMTAs* belonging to *Triticum aestivum* displayed lower expression under heat stress, indicating that *CAMTAs* played an essential role in the response to high temperatures [38]. Additionally, CAMTA1, CAMTA2, and CAMTA3 in *A. thaliana* could serve as regulators to repress the biosynthesis of salicylic acid and pipecolic acid in healthy plants, enhancing plant immunity [23,39]. When exposed to oxalic acid produced by *Sclerotinia sclerotiorum*, the expression of *BnCAMTA3A1* and *BnCAMTA3C1* from *Brassica napus* was elevated during the initial phase of infection. This indicates that *BnCAMTAs* might be involved in the pathogen defenses [40].

The *CAMTA* gene family plays a pivotal role in regulating plant growth and development, crucially mediating the plant response against a vast array of abiotic stresses encompassing heat, drought, cold, and biotic stresses mainly caused by bacteria, fungi, and viruses. The significance of the *CAMTA* family has been thoroughly documented in numerous plant species, with *Arabidopsis* as a prominent example [41]. However, the *CAMTA* gene family of *P. bournei* has not been reported. More importantly, there were relatively few studies that revealed the expression changes in *CAMTAs* under light stress; the response of *CAMTAs* to heat stress has also received attention. However, both of these environmental stresses are becoming increasingly apparent today. In this research, by applying bioinformatic analysis to *P. bournei*, we systematically identified the *PbCAMTA* gene family. The physicochemical properties, gene structure and chromosome distribution, and promoter cis-acting elements were investigated, and the evolutionary relationship of the *CAMTA* gene family was established. We analyzed the expression profile of *PbCAMTAs* under the conditions of drought, heat, and light stress genes based on qRT-PCR results. We hoped that our results provided data support and new insight for further understanding how *CAMTA* helps *P. bournei* cope with ever-worsening environmental conditions. This research offers a reference and foundation for delving deeper into the biological roles of the *PbCAMTA* gene family and the molecular pathways underlying their response to abiotic stresses.

## 2. Results

### 2.1. Identification and Characterization of PbCAMTA Genes

Based on assembled genome data, we identified 17 *CAMTAs* in *P. bournei*, which were renamed as *PbCAMTA1–17* according to the phylogenetic tree. To explore the characteristics of the *PbCAMTAs*, we analyzed their physical and chemical properties. Details of the derived *PbCAMTA* genes, such as the gene ID, number of amino acids, molecular weight, theoretical PI (isoelectric point), grand average of hydropathicity (GRAVY), instability index, and subcellular localizations of *PbCAMTAs*, are summarized in Table 1. The size of the *PbCAMTAs* ranged from 317 (*PbCAMTA12* and *PbCAMTA14*) to 1157 (*PbCAMTA3*) amino acids. Their molecular weights ranged from 35.63 kDa (*PbCAMTA1*2) to 129.66 kDa (PbCAMTA3). The predicted theoretical PI, ranging from 4.46 to 9.21, indicates that most *PbCAMTAs* can be viewed as acidic. The hydrophobicity indexes of the 17 members of this family were all negative, indicating that they can be classified as hydrophilic proteins. Eight of the family members had an instability index lower than 40, while nine of them were considered unstable. Notably, the subcellular localization prediction demonstrated that most *P. bournei CAMTA* families were located in the nucleus; two were in the cytoplasm (PbCAMTA1, 4), two were in the chloroplast (PbCAMTA11, 13), and two were in the plasma membrane (PbCAMTA15, 16).

### 2.2. Display of the Motif, Domain, and Gene Structure of PbCAMTA Gene Family

To further characterize the *P. bournei CAMTA* gene family, a phylogenetic tree was constructed with protein sequences of PbCAMTA (Figure 1). In total, 10 conserved motifs (Appendix A), domains, and exon–intron structures were demonstrated according to the phylogenetic tree. Most of the family members clustered in the same branches shared a similar motif and domain structure, while there was diversity between the *PbCAMTA* family. *PbCAMTA14*, *PbCAMTA15*, *PbCAMTA16*, and *PbCAMTA17*, which were clustered together, contained similar motif combinations, showing that they might execute similar functions. Furthermore, the ANKYR conserved domain was recognized in every *PbCAMTA*. The DHHC (Asp-His-His-Cys) domain was only found in *PbCAMTA14–16*. Therefore, it would be intriguing to further explore the multifaceted biological functions of the PbCAMTA proteins that possess these unique domains. Based on the exon–intron structure analysis, it can be concluded that 10 of the proteins preserved only the CDS (coding sequence) region, and the rest contained both the UTR (untranslated regions) region and the CDS region.

### 2.3. Phylogenetic Analysis of PbCAMTAs

To better understand and investigate the evolutionary relationship of the PbCAMTA transcriptional activator family, a phylogenetic tree was constructed based on full-length protein sequences of CAMTAs from five species: *Arabidopsis thaliana*, wheat (*Triticum aestivum*), soybean (*Glycine max*), tobacco (*Nicotiana tabacum*), and *P. bournei*. The number of CAMTA members in these species was 6, 17, 15, 19, and 17, respectively. A total of 74 CAMTA proteins were divided into 5 different subgroups (I–V) (Figure 2). As the phylogenetic analysis showed, the largest number of 26 CAMTA family members clustered in subgroup I, while there were only 4 *CAMTA* family genes in subgroup III. Among them, subgroup II comprised the largest number of seven members of the PbCAMTA family (PbCAMTA 12, PbCAMTA 13, PbCAMTA 10, PbCAMTA 11, PbCAMTA 1, PbCAMTA 2, and PbCAMTA 3), followed by subgroup V (PbCAMTA 14, PbCAMTA 15, PbCAMTA 16, and PbCAMTA17). In groups I, II, and IV, there were only two PbCAMTAs each (PbCAMTA 6 and PbCAMTA 7, PbCAMTA 8 and PbCAMTA 9, and PbCAMTA 4 and PbCAMTA 5, respectively).

Furthermore, CAMTAs that cluster together share high similarity in protein sequences, indicating the possibility of them sharing similar functions. In our study, CAMTAs of *P. bournei* generally clustered with CAMTAs from *Arabidopsis thaliana* and wheat (*Triticum aestivum*).

### 2.4. Chromosome Distribution of PbCAMTAs and Genomic Amplification in P. bournei

In total, there were 12 chromosomes in *P. bournei*, while the 17 *PbCAMTAs* were distributed stochastically on 8 chromosomes (Figure 3A). Among the chromosomes, including *PbCAMTA* genes, chromosome 9 contained the maximum number of *CAMTA* members: 4 (23.5%), namely, *PbCAMTA 4*, *5*, *8*, and *9*. Meanwhile, there was only one *PbCAMTA* gene located on each of chromosome 2 (*PbCAMTA 16*) and chromosome 10 (*PbCAMTA 3*). This uneven distribution pattern of *PbCAMTA* genes on chromosomes could be due to the genetic variability during the evolutionary process. Furthermore, eight pairs of collinear genes were discovered. Interestingly, *PbCAMTA 1*, *2*, and *3* on different chromosomes exhibited a collinear relationship. *PbCAMTA 15*, *16*, and *17* were also found to be homologous.

Gene families were mainly amplified through segmental replication, tandem replication, and whole-genome duplication in the process of biological evolution. In *P. bournei*, the whole genome might arise from transposition (dispersed, 38.95%) and WGD or segmental replication (whole-genome duplication, 29.29%) (Figure 3B). A Ka/Ks analysis of the whole genome showed that most of the genes experienced purified selection (Ka/Ks < 1); only a few were positively selected (Figure 3C). In our research, 8 pairs of *CAMTA* members in *P. bournei* experienced purified selection (Appendix A).

### 2.5. Syntenic Analysis of PbCAMTAs Genes

There were syntenic associations of the *CAMTA* genes between *P. bournei* and dicotyledons and monocotyledons (Figure 4). In the comparison between *P. bournei* and dicotyledonous species, including *A. thaliana*, *G. max*, and *N. tabacum*, four, seventeen, and four distinct syntenic associations were individually identified, respectively. Meanwhile, in monocotyledons, 23 homologous *CAMTA* gene pairs between *P. bournei* and *T. aestivum* were recognized. Moreover, 9 homologs and 13 collinear gene pairs were characterized in *O. sativa* and *Z. mays*, respectively. Chromosome 4, possessing *PbCAMTA7*, *11*, and *13*, was discovered to have the orthologous genes with all the other species. Notably, chromosome 1 (*PbCAMTA2*, *14*) and chromosome 5 (*PbCAMTA15*, *17*) exhibited a considerable amount of substantial orthologs between *P. bournei* and others. This result emphasized that duplication events were the primary force behind the expansion of the *CAMTA* gene family. It is worth noting that chromosome 2 and chromosome 9 showed only a collinear relationship between *P. bournei* and dicotyledon, while chromosomes 5 and 10 demonstrated only syntenic relationships between *P. bournei* and dicotyledon.

### 2.6. Cis-Regulatory Element Prediction of Promoters in 17 PbCAMTA Genes

Transcription factors typically operate by binding with specific cis-elements within the promoter regions of genes, thereby modulating the expression patterns of the genes located downstream (Appendix A). By screening the 2.0 kbp promoter sequences of *PbCAMTA* genes using the online software PlantCARE (https://bioinformatics.psb.ugent.be/webtools/plantcare/html/, accessed on 14 July 2024), multiple cis-regulators were identified. The identified cis-elements can be classified into four categories according to their distinct functions (Figure 5), including hormone responsiveness, stress signaling, light response, and growth and development. In our study, *CAMTAs* exhibited strong associations with hormone response elements, specifically abscisic acid responsiveness (ABRE), salicylic acid responsiveness (SARE), the MYB binding site involved in flavonoid biosynthesis (MYB-1), methyl jasmonate (MeJA), gibberellin responsiveness (GBRE), and auxin responsiveness (AuxRE). They also respond to external stresses such as drought (MYB-2, MYBHv1), cold stress (LTR), damage and defense, wound (WUN), heat stress (AT-rich), and anaerobic stress (ARE). More importantly, three light-related cis-elements were detected; they emerged as the most prevalent cis-elements in the *CAMTA* gene family. Light-responsive elements (Light) appeared in the promoter regions of every family gene in considerable accounts. In addition, a few *CAMTA* family genes contained cis-elements involved in plant growth and development, such as meristem-expression-related elements (MRE), zein metabolism regulation elements (Zein), circadian control elements (circadian), endosperm expression (Endosperm), seed-specific regulation (Seed), and cell cycle regulation elements (Cycle). In our study, *PbCAMTA1*, *2*, *4*, and *8*, which possess regulatory elements, were more likely to play pivotal roles when the plants encountered various abiotic stress. More attention should be paid to these family members.

### 2.7. Gene Ontology (GO) and KEGG Analysis in PbCAMTA

To further elucidate the functions of PbCAMTAs, gene ontology (GO) enrichment analysis was performed (Figure 6A). It encompasses three primary modules: molecular function, cellular components, and biological processes. As such, GO enrichment analysis furnishes us with directions and foundations for the subsequent exploration of the functions and mechanisms of PbCAMTAs. There is a possibility that PbCAMTAs could interact with small molecules, organic cyclic compounds, and other proteins. It is notable that PbCAMTAs were widely distributed in cells, which is in line with the prediction of subcellular localization. Additionally, PbCAMTAs potentially responded to cold stress. Furthermore, PbCAMTAs were found to participate in a pathway centered on membrane trafficking (Figure 6B).

### 2.8. Protein–Protein Interaction Network of PbCAMTA Proteins

After aligning the PbCAMTAs to AtCAMTAs, 16 homologous genes were characterized (Appendix A). Their interaction network was built via STRING and improved in Cytoscape (Figure 7). The results showed that CAMTA3, which was similar to PbCAMTA 6, 7, and 9, was highly connected with other proteins. In our study, CAMTA3 was found to be related to CAMTA 5, CAMTA 4, and CRK1 (cysteine-rich receptor-like kinases 1) [42]. These proteins were found to be involved in temperature responses. Interestingly, CAM7 (calmodulin 7) in Arabidopsis is highly important to enable A. thaliana to cope with light stress [43,44]. CAMTA2, which shared some similarities with PbCAMTA 8, interacted with 14 proteins. CAM7 might interact with proteins such as PHYB (phytochrome B) so as to enhance the ability of plants to tolerate light and other stresses [45]. Notably, based on the probable interaction between AtCAMTA and DREB (dehydration-responsive element-binding), HSF (heat-shock transcription factor) [46], PIA2 (phytochrome-interacting ankyrin-repeat protein 2) [47], and SCRM (SCREAM) [48], we could speculate that PbCAMTA also played a role in assisting plants to cope with abiotic stress [49]. AKR2A (ankyrin repeat-containing protein 2A) and EMB506 (embryo-defective 506), which align with PbCAMTA10 and PbCAMTA 13, individually interacted with proteins such as XBAT32 (XB3 ortholog 2 in Arabidopsis) and PIA2, respectively, indicating their potential role in the process of growth and development [50,51]. This understanding of the potential functional interplay between proteins aligning with PbCAMTAs and other important proteins paves the way for future functional validation and mechanistic studies.

### 2.9. Expression Patterns of PbCAMTA Genes in Response to Heat, Drought, and Light Treatments

Previous studies have thoroughly explored the central role of the *CAMTA* gene family in abiotic stress. We analyzed the expression of eight *PbCAMTAs* from five subgroups (I–IV) under three different abiotic stressors using quantitative real-time polymerase chain reaction (qRT-PCR). Stress treatments were as follows: polyethylene-glycol-induced drought (10% PEG-6000), heat (40 °C), and light (continuous 1200 µmol·mol^−1^·s^−1^). Primer sequences are listed in Table 2. EF1a was regarded as a reference sequence. The expression patterns of *PbCAMTAs* showed similar responses to these abiotic stressors, and some *PbCAMTAs* displayed pronounced upregulation or downregulation in response to specific stress conditions (Figure 8). Under drought stress, all *PbCAMTAs* were sharply upregulated and reached their peaks after four hours of PEG treatment; this was followed by downregulation. *PbCAMTA1* demonstrated the most remarkable rise (18.31 times), followed by *PbCAMTA 8*, *4*, *2*, and *6* (7.39 times average). This indicates that the *PbCAMTA* family has a strong response to drought tolerance. *PbCAMTAs* are dramatically expressed when exposed to high temperatures. Most *PbCAMTAs*, and specifically *PbCAMTA 6*, *12*, and *8*, reached their peak at 24 h (6.20, 4.98, and 4.85 times, respectively) and their lowest values after 8 h of heat stress. *PbCAMTA 16* and *17* reached their peaks at 4 h (on average, 3.49 times). This indicates that the *CAMTA* family is briefly suppressed at 8 h and then enhanced under heat conditions. After continuous light treatment, the *PbCAMTAs* exhibited a similar trend and were significantly expressed at 48 h. *PbCAMTA 12*, *6*, and *4* demonstrated drastic responses (7.92, 6.64, and 6.11 times, respectively). Detailed data are shown in Appendix A.

## 3. Discussion

*P. bournei* is one of the main sources of ‘golden-thread wood’. Not only does *P. bournei* have considerable practical value due to its hardness and corrosion resistance, but it also possesses enormous ornamental value. Due to increasingly extreme weather conditions, wild resources of *P. bournei* are scarce, and the size of the natural population is becoming smaller and smaller [52,53,54]. In this context, CAMTA is found to be an indispensable transcription factor that helps organisms overcome different constraints as well as optimize growth [13,14]. Studying the CAMTA proteins and understanding the mechanism of how they work in *P. bournei* can lay the groundwork and provide perspectives for future investigations.

In this study, we conducted a systematic analysis of the *CAMTA* gene family in *P. bournei*, identifying 17 genes as members of the family. Subsequently, basic biochemical properties (MW, PI, GRAVY, and subcellular localization) of the *PbCAMTAs* were predicted (Table 1). Compared to the *CAMTA* family identified in other species, the range of parameters of these properties in *PbCAMTAs* was relatively broad (specifically *PbCAMTA 9–17*), including for proteins containing only one (ANK) of the four domains within its classification. *PbCAMTA 1–8* shared general similarities with the *CAMTA* families of other species in terms of molecular weight (90–120 kDa), protein size (800–1000 aa), and isoelectric point (5–9), among other factors. Overall, these predictions offer valuable insights into the physical properties of the PbCAMTA proteins. The PbCAMTAs were mostly hydrophilic and were located in various positions, including the nucleus, cytoplasm, chloroplast, and membrane. The PbCAMTAs located in the nucleus were suspected to be related to the regulation of gene expression. The motif, domain composition, and gene structure of the PbCAMTA proteins were also investigated (Figure 1). The ANK domain, which is implicated in mediating protein–protein interactions, was identified in every member of the PbCAMTAs [37]. To gain complete insight into the *CAMTA* gene family, a three-dimensional structure was also predicted (Appendix A).

To gain deeper insights into the evolutionary relationships among *A. thaliana*, wheat (*Triticum aestivum*), soybean (*Glycine max*), tobacco (*Nicotiana tabacum)*, and *P. bournei*, a phylogenetic tree of the CAMTAs was constructed. The 74 CAMTA proteins under investigation were categorized into five distinct subgroups based on the established classification criteria (Figure 2). Given the likelihood that genes clustered within the same subgroup may share similar functional properties, the potential functions of the PbCAMTAs clustered with studied CAMTAs could be deduced. Interestingly, PbCAMTAs were more frequently clustered with *A. thaliana* and *T. aestivum*. Therefore, we hypothesized that the *PbCAMTA* genes had a similar function to their counterparts in *Arabidopsis* and *T. aestivum*. In subgroup 1, PbCAMTA6 and PbCAMTA7, which clustered with AtCAMTA1, probably regulate drought recovery by influencing the expression of AP2-EREBP transcription factors and modulating the plant’s response to abscisic acid [35]. Remarkably, the expression of *TaCAMTA1b-B.1* transgenic lines showed significant upregulation when exposed to drought stress, indicating that *TaCAMTA1b-B.1* is highly important for plants’ ability to cope with water scarcity during the seedling stage [31]; this suggests the function of PbCAMTAs in subgroup I. In subgroup IV, the overexpression of GmCAMTA12, which clustered with PbCAMTA4 and 5, enabled soybeans to resist drought stress by upregulating drought tolerance genes [36]. In subgroup I, AtCAMTA3, also known as SR1, emerged as a multifaceted regulator that is involved in crucial signaling pathways. It was recognized to play a pivotal role in orchestrating ethylene and abscisic acid signaling, as well as being intricately involved in the interpretation of light signals [55]. The *GmCAMTA2* and *GmCAMTA8* genes were thought to be subject to circadian rhythms, serving as inhibitory factors in development and drought stress responses. Moreover, we conducted a collinear analysis between *P. bournei* and other dicotyledonous and monocotyledonous plants (Figure 4). The number of homologous gene pairs found between *P. bournei* and *T. aestivum* was the highest, followed by *G. max*. Furthermore, more collinear gene pairs were identified between *P. bournei* and monocots than with dicotyledons, hinting that the differentiation of the *CAMTA* gene family occurred in basal angiosperm, which was consistent with previous studies [14]. Previous studies have reported that, during the evolutionary process, gene families are mainly generated and maintained through tandem duplication and segmental genome replication events [56]. In the inter-species collinearity analysis, whole-genome duplication (WGD) stands as the primary catalyst driving the expansion of the *PbCAMTA* family. This large-scale genomic event facilitated the proliferation of *PbCAMTA* members, contributing significantly to the genetic diversity and adaptability of the family in response to diverse environmental stimuli (Figure 3B). Eight collinear gene pairs discovered in *P. bournei* might have been generated as a result of chromosomal segmental duplication, while *PbCAMTA 11*, *12*, and *14* were generated by transposition. Additionally, the assessment of evolutionary pressures on these protein-coding genes could be achieved by examining the rates of non-synonymous (Ka) and synonymous (Ks) substitutions and their ratio (Ka/Ks) [57]. The Ka/Ks ratio of all the *PbCAMTA* gene pairs was found to be less than 1, indicating that these gene pairs have likely been subjected to significant purifying selection pressure with constrained functional variation during evolution (Figure 3C).

In line with the environmental changes and the locations of *P. bournei*, there is an urgent need to identify the resistance mechanisms of *PbCAMTAs* that respond to drought, heat, and light. Notably, *PbCAMTAs* could also respond to environmental stresses such as WUN and ARE. The growth and development of plants were also mediated by *PbCAMTAs*. The results are consistent with previous reports [58,59]. *PbCAMTA 1*, *4*, and *8* are the genes most strongly related to drought. Subsequently, the protein–protein interaction network was predicted using *Arabidopsis* as a reference, and the PbCAMTA6, 7, and 8 with homology to CAMTA3 could be speculated to have similar functions and protein interaction networks in *P. bournei*. They may interact with proteins homologous to CAM7, which is related to temperature response. AtCAMTA3 has been preliminarily studied, and its role in responding to stress has been indicated [60,61]. Based on our previous study, we chose eight *PbCAMTAs* for an extensive examination of their expression patterns. The expression patterns of *PbCAMTA* genes under drought conditions, heat stress, and light intensities were analyzed utilizing qRT-PCR. All *PbCAMTAs* were upregulated when exposed to multiple stresses, indicating their potential role in enhancing the stress resistance capabilities of *P. bournei*. This suggests that *PbCAMTA* acts as a transcriptional activator to regulate the plant’s adaptive responses to environmental challenges. Accordingly, *PbCAMTA 4* and *PbCAMTA 6* exhibited the most obvious changes when encountering three kinds of stress. *PbCAMTA 1* showed the most evident upregulation under drought stress. Overall, these findings reveal that the *CAMTA* gene family plays an essential role during the development of *P. bournei*, allowing *it* to gain resilience to various environmental stresses. In general, the identification and characterization functions of *PbCAMTAs* in response to drought, heat, and light have provided promising genetic strategies for improving the stress tolerance in *P. bournei*.

CAMTA is a crucial transcription factor, endowing plants with the ability to cope with various stresses. In previous studies, we found that abiotic stresses such as drought, heat, or light stress stimulate Ca^2+^ channels (TPCs, CNGCs, MS) to transport Ca^2+^ [62]. With increasing Ca^2+^ concentrations in the cytoplasm, numerous physiological activities within cells are regulated; we focused most on the changes in cellular metabolic activities induced by PbCAMTA for environmental adaptation. Under the conditions of a water deficit from either drought or high salinity, the immediate response of plants is to close the stomata to diminish water loss and accumulate compatible solutes such as proline and glycine betaine to adjust the osmotic balance [63,64,65]. Notably, sugars that can be accumulated through the action of dehydration-responsive element-binding proteins (DREB) play a significant role in osmotic adjustments [66,67]. The protein interaction analysis showed that PbCAMTAs might interact with DREB so as to adapt to drought stress. Plants adopt various strategies to adapt to high temperatures, the most common of which include maintaining membrane stability, eliminating reactive oxygen species (ROS), producing antioxidants, and regulating the level of compatible solutes [68]. The induction of CPK signaling cascades, along with chaperone signaling and transcriptional activation, is a common strategy utilized by plants to mitigate heat stress [69]. Continuous light stress causes protection mechanisms, including scavenging ROS in the chloroplast and stomatal movement [70]. Studies have indicated that PHYB is a light and temperature sensor; it can regulate the expression of target genes under light conditions [71,72,73]. In summary, scavenging ROS and maintaining the osmotic balance are key to equipping plants with the resilience to thrive under stressful conditions [74,75]. In our study of cis-elements and protein–protein interactions (Figure 5 and Figure 7), we found a potential relationship between these critical responding proteins and PbCAMTA, suggesting that PbCAMTA plays an indispensable role in defending against environmental stress by interacting with or regulating these proteins (Figure 9). The mechanism was our speculation based on studies of *PbCAMTA* and other research concerning plants’ common responses to stresses. Further experiments are needed to verify whether *PbCAMTAs* truly help plants to confront stress challenges by interacting with or regulating the expression of the proteins mentioned above.

## 4. Materials and Methods

### 4.1. Identification and Analysis of CAMTA Gene Families in P. bournei

The genome sequence and annotation profile of *P. bournei* were obtained from the Sequence Archive of China National GeneBank Database (https://db.cngb.org/search/project/CNP0002030/, accessed on 30 June 2024) [76]. The conserved domains of the CAMTA were commonly comprised of the CG-1 DNA-binding domain (Pfam03859), IQ motifs (Pfam00612), ANK anchor protein repeats (Pfam12796), and TIG domain-binding non-specific DNA (PFam01833). Sequences of these domains were accessed through the Pfam protein family database (http://pfam.xfam.org/, accessed on 30 June 2024). Hmmer search (3.0) was used to identify genes with the target domain. Later, the amino acid sequences of *A. thaliana* CAMTA were obtained from the TAIR database (https://www.arabidopsis.org/, accessed on 28 June 2024). BLASTp at the National Center for Biotechnology Information (NCBI) was used to align the proteins of *P. bournei* with AtCAMTAs, and results with high confidence levels were reserved. By combining the results from these two methods, the *PbCAMTAs* were identified. The *PbCAMTAs* were subsequently checked using the NCBI-CDD search tool (https://www.ncbi.nlm.nih.gov/Structure/bwrpsb/bwrpsb.cgi, accessed on 28 June 2024) and SMART network database (http://smart.embl-heidelberg.de/, accessed on 28 June 2024). Finally, 17 genes were identified as *CAMTA* family members in *P. bournei* and were renamed *PbCAMTA1–17*.

### 4.2. Physical and Chemical Characteristics of P. bournei

The physical and chemical characteristics of the *PbCAMTA* family members were analyzed using the ProParam module of Expasy online software (https://www.expasy.org/, accessed on 2 July 2024) and WoLF PSORT online software (https://www.genscript.com.cn/, accessed on 2 July 2024) to predict the subcellular localization.

### 4.3. Analysis of the Phylogenetic Tree and Motif, Domain, and Gene Structures of PbCAMTAs

MEGA (version 11.0.10) [77] was used to analyze the evolutionary relationships in the PbCAMTAs. The evolutionary history was inferred using the maximum likelihood method and the Poisson correction model. A discrete Gamma distribution was used to model differences in the evolutionary rate among sites (+*G*, parameter = 6.33). MEME (https://meme-suite.org/meme/tools/meme, accessed on 4 July 2024) was used to identify the conserved motifs in the *PbCAMTA* family. The number of motifs to be found was set to 10 [78]. Other parameters were kept as the default. The intron–extron gene structure was displayed using the online software GSDS (https://gsds.gao-lab.org/, accessed on 4 July 2024). TBtools were used to visualize the results of the above analysis and the results of the CDD search.

### 4.4. Phylogenetic Analysis between PbCAMTAs and CAMTAs of Other Species

The CAMTA protein sequences of *Arabidopsis thaliana* (https://www.arabidopsis.org/, accessed on 1 July 2024) and *Triticum aestivum* were obtained from the Ensemble Plant database (Ensembl Plants, accessed on 29 June 2024). The genome and annotation data of *Glycine max* (SoyBase.org) [26] and *Nicotiana tabacum* [21] *(*https://solgenomics.sgn.cornell.edu/organism/Nicotiana_tabacum/genome accessed on 4 July 2024) (Appendix A). Sequences from the chosen species were aligned using ClustalW (Multiple Sequence Alignment—CLUSTALW (genome.jp)). The phylogenetic tree was constructed with MEGA software version 10.2 using the neighbor-joining method and the Poisson correction model. Specific parameters were set as the Jones–Taylor–Thornton (JTT) model and 1000 bootstrap replications (with a bootstrap cutoff of 50). Furthermore, the phylogenetic tree was visually enhanced using the iTOL website (Interactive Tree of Life, embl.de).

### 4.5. Collinearity Analysis and Duplication Events Analysis of CAMTA

MCScanX (https://github.com/wyp1125/MCScanX/, accessed on 10 July 2024) was used to analyze the syntenic relationships between *PbCAMTA* and *CAMTA* genes from *Arabidopsis thaliana*, *Glycine max*, *Nicotiana tabacum*, *Triticum aestivum*, *Oryza sativa*, and *Zea mays* [79]. The segmental and tandem duplication events that occurred in *PbCAMTA* genes were also determined using MCScanX. TBtools-II v2.10 [80] software was employed to visualize the distribution and collinearity of the *PbCAMTAs*. R program (4.4.1, r-project.org) was used for the statistical computing, and R package ggplot2 was used for the visualization.

### 4.6. GO and KEGG Analysis

The GO and KEGG annotation of CAMTA proteins was executed using the eggNOG-MAPPER platform (http://eggnog-mapper.embl.de/, accessed on 14 July 2024) [81]. The numbers of the PbCAMTA proteins were annotated and grouped into three categories. The GO and KEGG annotation results were visualized using TBtools.

### 4.7. Protein–Protein Interaction Network of PbCAMTAs

*P. bournei* homologs of the *Arabidopsis* proteins displayed in the interaction network were identified using the BLASTp program. The CAMTA protein sequences were uploaded and analyzed utilizing the STRING database (https://cn.string-db.org/, accessed on 12 July 2024). The prediction of relationships among significant proteins was undertaken based on the analysis of protein interactions observed in *Arabidopsis*. Cytoscape (V3.10.2) was used to visualize and refine the resulting network [82].

### 4.8. Abiotic Stress Treatment

Seedlings were meticulously selected from 1-year-old *P. bournei* specimens. Before the plants were exposed to stress, the soil used to plant the seedlings was prepared using peat soil, humus soil, sandy soil, perlite, etc. in a ratio of 5:2:2:1. The soil organic matter content ranged from 2.57% to 6.07%. The growth area experienced an average annual temperature of 16–20 °C, accompanied by annual precipitation ranging from 900 mm to 2100 mm and an approximate annual relative humidity of 77%. In our experiment, the drought and heat treatment groups were sampled at 0, 4, 8, 12, and 24 h, while the light treatment group was sampled at 0, 24, 48, and 72 h. The seedlings collected at 0 h were considered the control group. Each stress treatment comprised 3 individuals (treated for 0 h). The drought and heat stress treatment groups had 4 individuals, respectively, while light stress comprised 3. Each group underwent tailored experimental conditions. During the stress treatment, the setting parameters of the artificial climate chambers were as follows: light cycle 12 h/d, LED lights used for lighting, photosynthetically active radiation set to 1200 µmol·mol^−1^·s^−1^, temperature 25 °C. The experimental protocol entailed simulating drought stress, where the treatment group was transplanted into a beaker containing 10% PEG 6000 of ¼-strength Hogland solution, an approach designed to mimic drought conditions. For the temperature treatments, individuals were incubated at an elevated temperature of 40 °C. As for the light treatments, the corresponding treatment groups were under the same light intensity (1200 µmol·mol^−1^·s^−1^) but exposed for 24, 48, and 72 h, and the control group was sampled after 12 h of dark treatment from half of the light cycle. After treatment, leaf specimens were promptly harvested and preserved in liquid nitrogen at −80 °C, facilitating the subsequent RNA extraction.

### 4.9. RNA Extraction and Statistical Analysis

Utilizing the RNA Extraction Kit from Omega Bio-Tek (Shanghai, China), total RNA was extracted from both the control and stress-treated samples. Adhering strictly to the manufacturer’s guidelines, the EasyScript One-Step gDNA Removal and cDNA Synthesis SuperMix from Transgen (Beijing, China) was then employed to synthesize cDNA from the extracted RNA. Quantitative RT-PCR was subsequently performed using TransStart top green qPCR SuperMix (Transgen, Beijing, China). The mixture solution of the qRT-PCR reaction is composed of 1 µL of cDNA, 2 µL specific primers, 10 µL of SYBR Premix Ex TaqTM II, and 7 µL of ddH_2_O. The qRT-PCR reaction process was as follows: degeneration at 95 °C for 30 s, then 40 cycles of denaturation at 95 °C for 5 s, 60 °C for 30 s, 95 °C for 5 s, 60 °C for 60 s, and 50 °C for 30 s. The relative expression of the *PbCAMTA* genes was calculated using the delta–delta Ct method, and one-way ANOVA analysis with a confidence level of 95% and Duncan multiple comparison tests were performed using GraphPad Prism9.0 software (https://www.graphpad.com/) [83,84]. To ensure robustness, all quantitative PCRs were conducted with three biological repeats and three technical replicates.

## Figures and Tables

**Figure 1 ijms-25-09767-f001:**
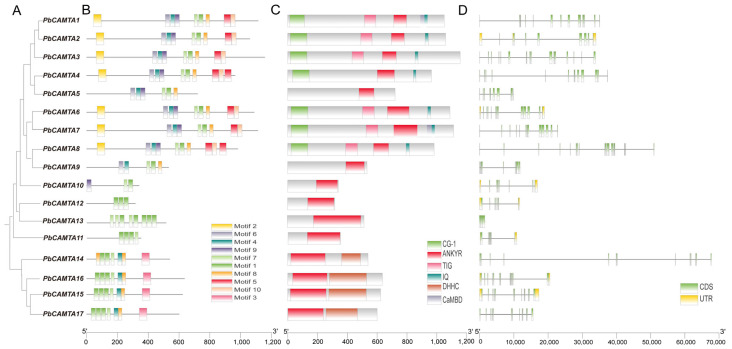
Protein motifs, domains, and structures of the *CAMTA* gene family in *P. bournei*. (**A**) The *DlGATA* phylogenetic tree was constructed using MEGAX software version 10.2, and the results were consistent when using the maximum similarity method with 1000 repetitions. (**B**) Protein motifs in PbCAMTA members. The colorful boxes demarcate different motifs. (**C**) Demonstration of conserved domains. (**D**) Gene structures of the *PbCAMTA* gene family. CDS regions are represented by green rectangles, while UTR is shown in yellow. Black lines denote introns.

**Figure 2 ijms-25-09767-f002:**
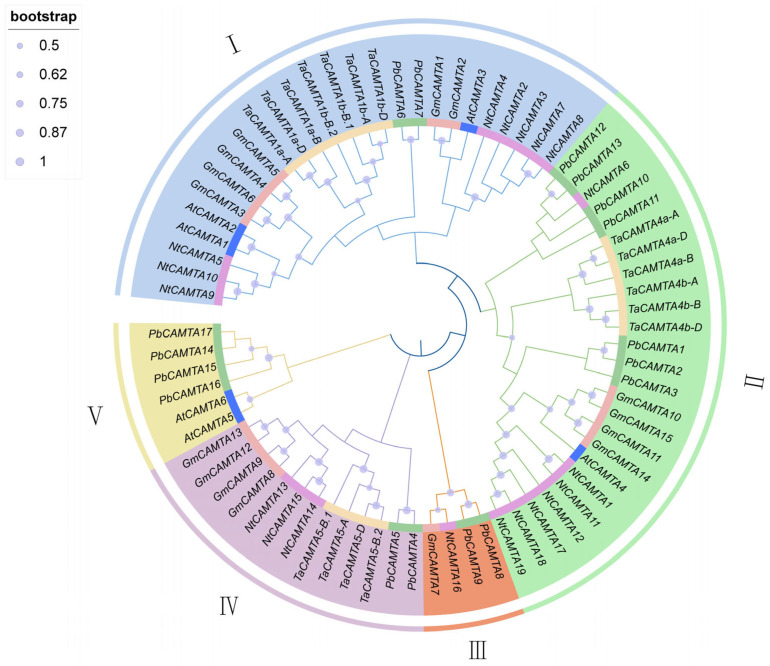
A phylogenetic analysis of CAMTA proteins from *P. bournei* (Pb), *Arabidopsis thaliana* (At), *Triticum aestivum (Ta*), *Glycine max (Gm*), and *Nicotiana tabacum* (*Nt*) was carried out using the neighbor-joining method (setting the maximal likelihood method and bootstrapping 1000). Different species are shown in different colors in the inner circle. The five subgroups (Group I–V) of CAMTA proteins are represented by different colors in the outer circle.

**Figure 3 ijms-25-09767-f003:**
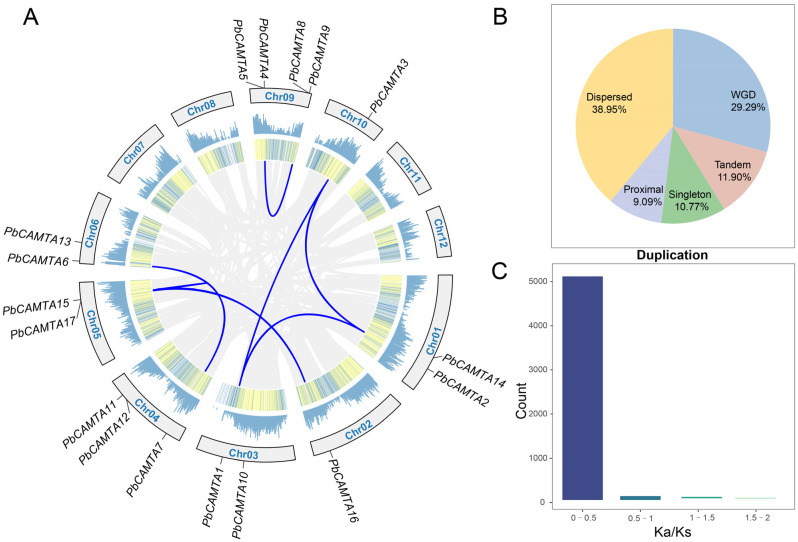
The distribution of *PbCAMTAs* in 12 chromosomes and genomic amplification. (**A**) The density of *CAMTA* genes is presented along with each chromosome in the line plot and heatmap. Gene pairs originating from *PbCAMTA* are linked by blue lines. (**B**) Statistics of the duplication events in the whole of *P. bournei*. (**C**) Ka/Ks values are calculated and sorted into four levels (0–0.5, 0.5–1, 1–1.5, and 1.5–2).

**Figure 4 ijms-25-09767-f004:**
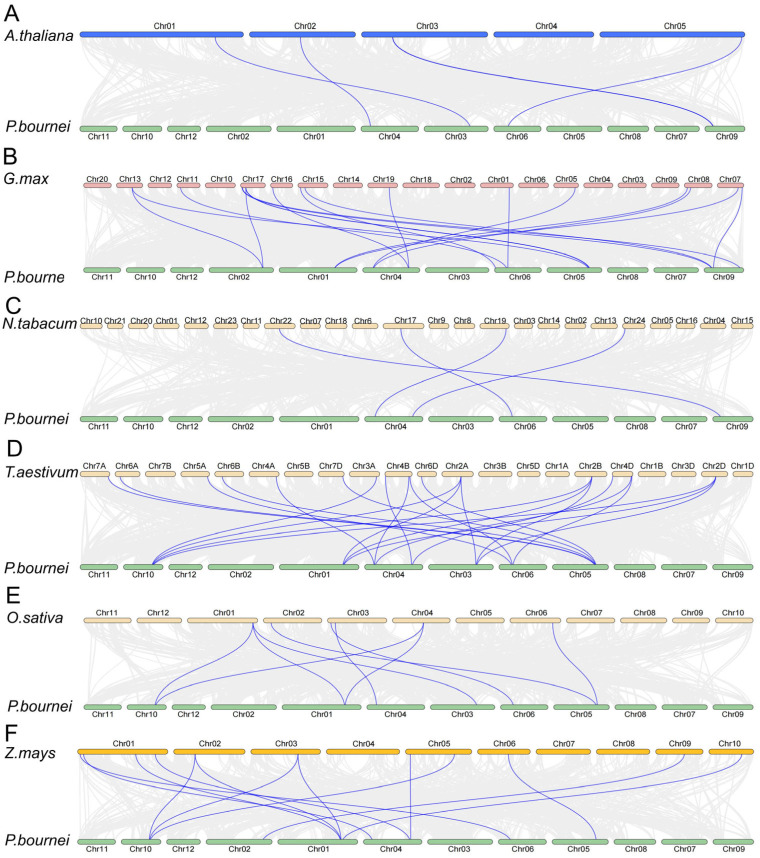
Comparative synteny analysis of *CAMTA* genes across various plant species, including chromosomes from *P. bournei*, as well as those of *A. thaliana* (**A**), *Glycine max* (**B**), *Nicotiana tabacum* (**C**), *Triticum aestivum* (**D**), *Oryza sativa* (**E**), and *Zea mays* (**F**) chromosomes. The background’s gray lines emphasize the syntenic *CAMTA* gene pairs, while the collinearity of *PbCAMTA* and the other six species are highlighted with blue lines. The chromosome number for each species is clearly indicated at the top or bottom of each respective chromosome depiction.

**Figure 5 ijms-25-09767-f005:**
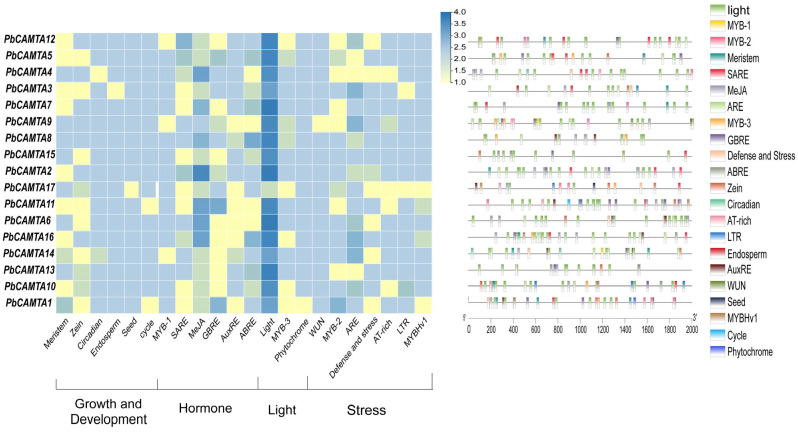
Schematic distribution and visual overview of the localization of *cis*-acting elements in each *PbCAMTA*.

**Figure 6 ijms-25-09767-f006:**
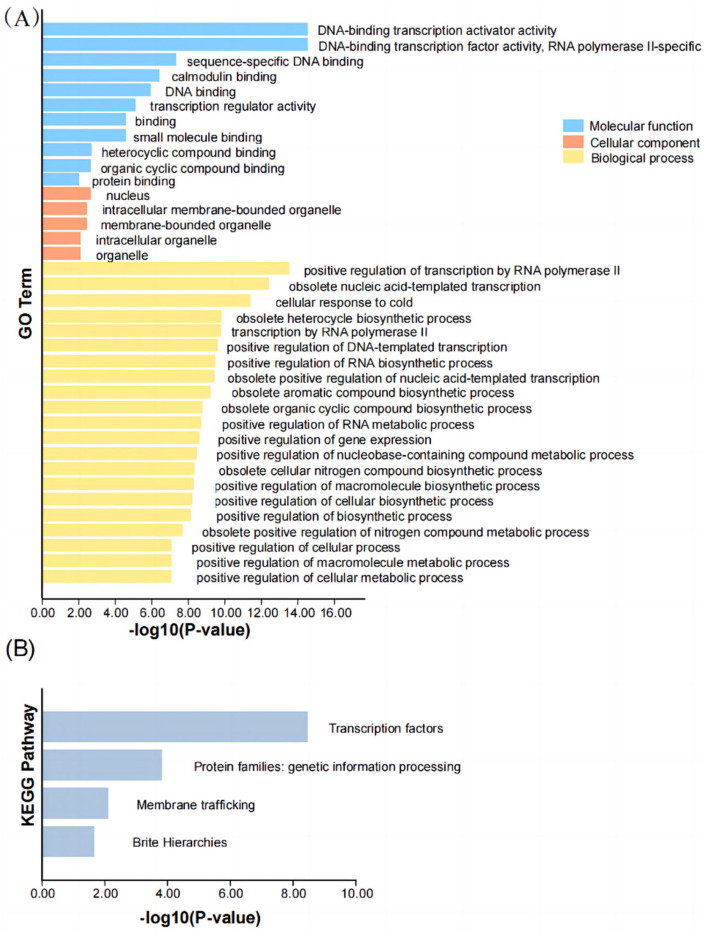
GO enrichment (**A**) and KEGG analysis (**B**) of *PbCAMTA* genes.

**Figure 7 ijms-25-09767-f007:**
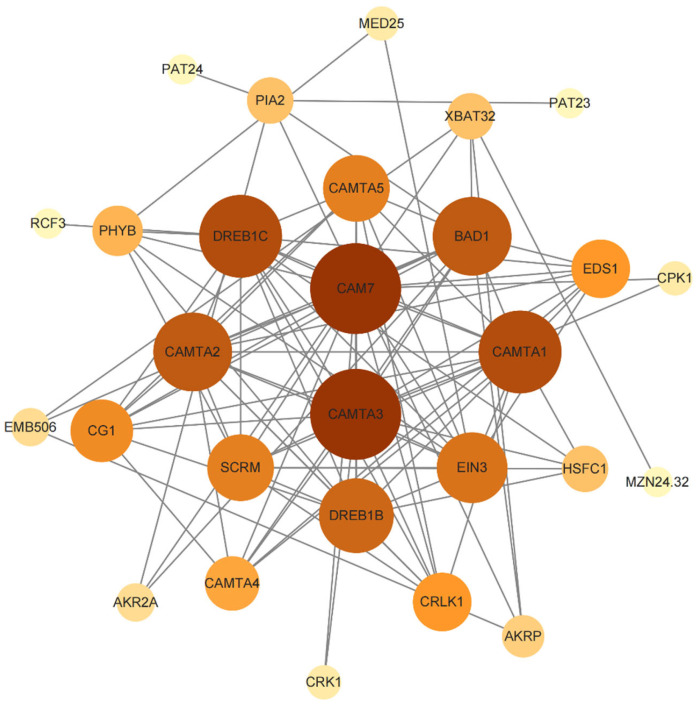
Protein–protein interaction network produced by STRING. The more proteins a protein is correlated with, the bigger the circle is and the deeper the color.

**Figure 8 ijms-25-09767-f008:**
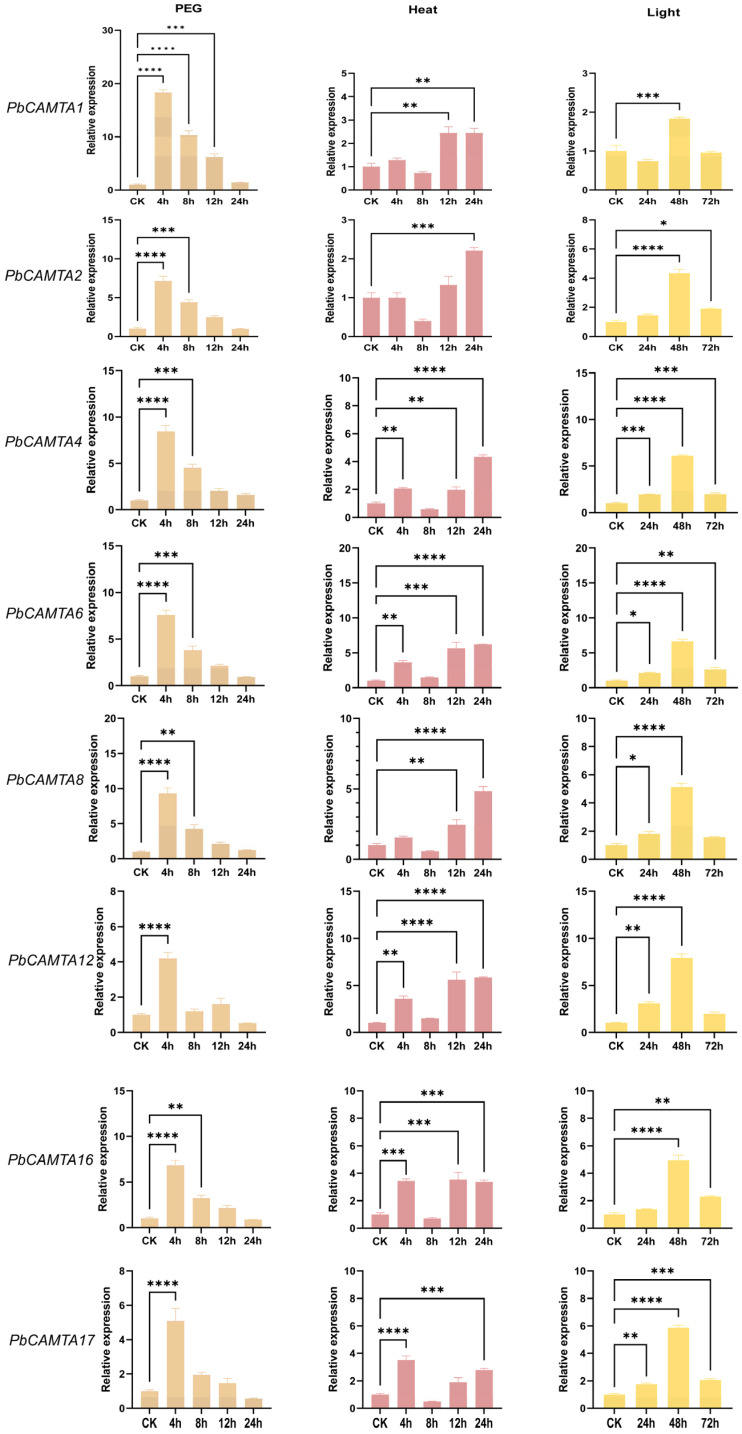
Expression profiles of *PbCAMTA* genes under various stresses. Error bars indicate the standard deviation (SD). The statistical analysis employed one-way ANOVA to discern significant differences, with the number of ‘*’s representing the level of significant differences (* *p* ≤ 0.05; ** *p* ≤ 0.005; *** *p* ≤ 0.0005; **** *p* ≤ 0.0001).

**Figure 9 ijms-25-09767-f009:**
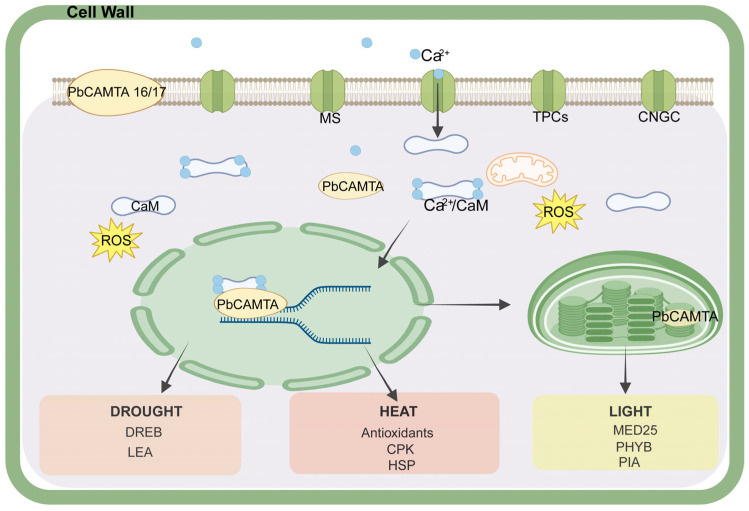
Predicted mechanisms of CAMTAs that enable plants to cope with severe environmental stresses. The colored squares indicate proteins responding to different stresses, where orange stands for drought stress, pink stands for heat stress, and yellow stands for light stress.

**Table 1 ijms-25-09767-t001:** Physical and chemical characteristics of the *P. bournei CAMTA*.

Gene Name	Gene ID	Size/aa	Molecular Weight/kDa	Theoretical PI	Grand Average of Hydropathicity	Instability Index	Subcellular Localization
*PbCAMTA1*	OF24973	1050	117.00	5.95	−0.538	40.02	cytoplasm
*PbCAMTA2*	OF11451	1059	117.88	5.39	−0.464	42.68	nucleus
*PbCAMTA3*	OF06278	1157	129.66	5.83	−0.553	44.52	nucleus
*PbCAMTA4*	OF03012	964	108.86	6.59	−0.377	82.89	cytoplasm
*PbCAMTA5*	OF03004	720	79.77	5.27	−0.255	34.69	nucleus
*PbCAMTA6*	OF19359	1088	121.79	5.79	−0.525	44.20	nucleus
*PbCAMTA7*	OF07073	1112	124.82	5.82	−0.561	45.34	nucleus
*PbCAMTA8*	OF07605	982	110.74	8.42	−0.518	36.32	nucleus
*PbCAMTA9*	OF07600	532	36.37	5.68	−0.520	47.73	nucleus
*PbCAMTA10*	OF24143	340	55.07	4.46	−0.564	39.56	nucleus
*PbCAMTA11*	OF17261	350	38.75	6.04	−0.498	60.49	chloroplast
*PbCAMTA12*	OF00973	317	35.63	9.21	−0.5115	40.97	nucleus
*PbCAMTA13*	OF21192	514	55.07	7.02	−0.058	39.45	chloroplast
*PbCAMTA14*	OF20820	317	59.31	6.33	−0.066	33.92	nucleus
*PbCAMTA15*	OF09478	539	69.12	6.37	−0.207	32.58	Plasma memberane
*PbCAMTA16*	OF20135	635	70.07	6.59	−0.219	31.69	Plasma membrane
*PbCAMTA17*	OF14546	600	66.96	6.32	−0.002	28.67	nucleus

**Table 2 ijms-25-09767-t002:** Primer sequences used for qRT-PCR.

Gene Name	Forward	Reverse
*PbCAMTA1*	GAACCATGCCCATCGATCA	GATATTGCTGCCGTGCCTTG
*PbCAMTA2*	GGTAGAACTGTTGGGGAGGC	TCCAATAGCAGCGCCTTTGA
*PbCAMTA4*	AGGTACTTCGCTGGTTGGTG	GCCTCCAAAAGATCTGCCCA
*PbCAMTA6*	TGCTGGTTACCTTGCGGAAT	CGCTCTGCAACTGTTTGGAC
*PbCAMTA8*	CCGCCATCGAATGCATCAAG	CTGTAGCATCCGTCAGTCCC
*PbCAMTA12*	ACGCATACCTAATCTGGCCG	ACATTGTGTTTGCGCAGCTT
*PbCAMTA16*	AACGGCCATGTCTGTGTTGA	GGAACCCATCAGACTTGCCA
*PbCAMTA17*	CTGCCCTATTGGCTGGGAAT	GCTCAACACAACGATCGCAA
*EF1a*	GCTCCTGGTCACCGTGAC	TCACGAGTTTGCCCGTCC

## Data Availability

Data is contained within the article and Appendix A.

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
