# Peer review of "The Vital Role of the *CAMTA* Gene Family in *Phoebe bournei* in Response to Drought, Heat, and Light Stress"

_ijms, 2024, doi:10.3390/ijms25189767_

Round 1

Reviewer 1 Report

Comments and Suggestions for Authors

My main objections are as follows:

1. Quality of English is not good enough.

2. The text lacks references to the supplementary files.

3. In some figures the descriptions are illegible.

4. Paragraph 2.9 needs to be rewritten as it does not provide any relevant information.

5. Discussion lacks references to the relevant figures.

6. There are far too many commonly known statements in the discussion section.

7. The paragraph 4.7 is not clear and confusing.

I have highlighted other comments in the attached file.

Comments on the Quality of English Language

As I have indicated in the review form and in many places in the manuscript, moderate English editing is required.

Reviewer 2 Report

Comments and Suggestions for Authors

Comments:
The study contributes to the scientific knowledge for understanding the role of CAMTA gene family in modulating abiotic stress tolerance in Phoebe bournei. While the concept and approach of the study have merits, the manuscript needs substantial improvement, especially in describing sufficient details of Materials Methods, and Results. The study has no physiological and minimal molecular characterization of plants under different abiotic stress tolerance phenotypes observed in this study. I have provided comments and suggestions throughout the manuscript. My comments are summarized below:

Title: Identified needs to be removed and the title should be changed based on the role of the CAMTA gene family in abiotic stress tolerance.

Abstract:

1. Line 22: various abiotic stresses. 

2. Line 26: could be varied depending on their subcellular localization. 

Keywords: Consider selecting the keywords that are not reported in the title.

Introduction:

1.      Line 45: various factors, authors need to specify if they mean abiotic, biotic, or both.

2.      Line 76-78: The scientific name of the plant should be provided in the parenthesis in the first place and then use the common name after that. Line 76 says Nicotiana tabacum and no common name similarly line 77 Oryza sativa and no common name.

3.      Line 95-98: Rephrase it for easy understanding of readers.

4.      Line 99: Authors need to explain the rationale behind this work and what research gaps it addresses. I suggest moving lines 326-332 to line 99.

Methods:

1.      Section 4.2: Details about how the evolutionary tree was generated is missing in this section as the title reads analysis of evolutionary tree.

2.      Section 4.7: Details about the experimental setup, design, and conditions are missing. The authors need to provide details on where the experiment was conducted, how plants were grown and what were the growth conditions, what was the age of plants when exposed to different abiotic stress conditions, and the number of replicates.

3.      Line 522: Why did the authors choose to have 30 individuals in control and 3 individuals in stressful conditions?

Results:

1.      Line 112: polygenic tree?

2.      Line 118: The molecular weight should be converted to kDa both in text and table.

3.      Section 2.1: Authors should include the information about solubility of proteins and whether they are monomeric or multimeric.

4.      Line 293: Replace researches with studies.

5.      Line 296: replace courting with under three different abiotic stressors.

6.      Line 297-311: Authors need to mention and cite figure of each gene in the sequence in it listed in Figure 9 and mention the fold difference relative to control in the parenthesis. Please see the following article for help:

Rathor, P., Borza, T., Stone, S., Tonon, T., Yurgel, S., Potin, P., & Prithiviraj, B. (2021). A novel protein from Ectocarpus sp. improves salinity and high temperature stress tolerance in Arabidopsis thaliana. International Journal of Molecular Sciences22(4), 1971.

7.      Section 2.10 and Figure 9: Why authors did not include the control plants here (plants growing under normal conditions) along with the plants that were subjected to different abiotic stressors? Authors need to include untreated control plants for each condition and time point for the gene expression analysis. Why Ck is only at one-time point when you are studying the expression pattern for 24 hours. Each time point should have an untreated control along with it to say that changes in expression are due to imposed conditions and not due to developmental conditions.

8.      Line 430: adjust osmotic, this sentence is incomplete.

Minor comments:
1. Authors should be consistent in italicizing gene names as Table 1, line 222, figure 9 gene names are not italisized. It should be italicized through the MS.
2. Authors should carefully read the MS and needs to be edited by native English speaker as there are several grammatical errors and at several places. For example, line 58-59, line 153, line 238, line 328-330, line 359 and multiple other places which all can not be listed.

3. Authors should italicize the scientific names as multiple places there are mentioned without italicizing it. For example: line 212-213, Arabidopsis thaliana, Glycine max, Nicotiana tabacum; line 274, P. bournei, Arabidopsis thaliana; line 413 and multiple other places.

Comments on the Quality of English Language

Authors should carefully read the MS and needs to be edited by native English speaker as there are several grammatical errors and at several places. For example, line 58-59, line 153, line 238, line 328-330, line 359 and multiple other places which all can not be listed. 

Reviewer 3 Report

Comments and Suggestions for Authors

The presented manuscript had the goal to uncover and begin the characterization of CAMTA family genes in the ornamental and wood source species Phoebe bournei. The authors successfully identified 17 of these genes in the current assembly of the P. bournei genome, and have sufficient data to demonstrate their conservation in comparison to CAMTAs of other species. Nonetheless, some experimental details are unclear and some statements are overconfident, demanding more attention. Below are comments that will hopefully aid in improving the manuscript prior to publication.

1. Line 66: There is a repetition of the word "Generally" in this line. Please revise for clarity.

2. Lines 79-80: The referenced study (reference 29) identified CAMTAs in five specific Solanaceae species: Solanum melongena, Solanum lycopersicum, Solanum pennellii, Capsicum annuum, and Lycium barbarum, with a focus on S. melongena. To avoid confusion, I suggest changing "in Solanaceae" to "in five different Solanaceae species" or specifying "Solanum melongena" to accurately reflect the study's content. This will help clarify that not all Solanaceae species were examined, such as Nicotiana tabacum, mentioned in line 76, which was not part of the study in reference 29.

3. Lines 85-98: When discussing the functions of CAMTAs, it would be helpful to briefly mention the species along with the gene names. For instance, "GmCAMTA12 is an important transcription factor in soybean that can be exploited..." or "AtCAMTA6 in Arabidopsis thaliana modulates...". This would make it easier for readers unfamiliar with the gene abbreviations to follow the text.

4. Line 113: The phrase "on Expasy" can be omitted, as it should be included in the methods section.

5. Line 122: Consider revising "are considered instability" to "are considered unstable."

6. Lines 130-132: It would be clearer to describe the sequence analysis software and method only in the methods section to avoid redundancy.

7. Lines 139-140: It was unclear to me which genes belong to each subfamily. Is this information provided in Figure 1?

8. Figure 1: The text in Figure 1 is difficult to read. Please consider increasing the font size for clarity. Additionally, indicate in the legend that the colored boxes in the phylogenetic tree represent MEME motifs in panel A (this is currently in panel B). Placing the color legend next to the tree in panel A would help avoid confusion with the colored boxes in panel D (representing UTR and CDS). The color legend in panel D is also incorrect—yellow represents UTRs and green represents CDS.

9. Lines 153-158 and Figure 2: The statements regarding the similarity of three-dimensional structures would be strengthened by performing structural alignment. I recommend using tools such as MatchMaker (doi:10.1186/1471-2105-7-339) from ChimeraX (doi:10.1002/pro.3943) and/or TM-align (doi:10.1093/nar/gki524) to calculate pairwise structural alignment scores. This would add more experimental confidence to the claims in this section.

10. Lines 173-175: I suggest revising the phrasing for clarity: "In groups I, II, and IV, there are only two PbCAMTAs each (PbCAMTA 6 and PbCAMTA 7, PbCAMTA 8 and PbCAMTA 9, PbCAMTA 4 and PbCAMTA 5, respectively)."

11. Figure 4 (B and C): The font size and resolution need improvement, as the text is difficult to read.

12. Line 211: There may be a missing "and" before "monocotyledons."

13.  Lines 212-213: The species names should be italicized.

14. Section 2.8: Some of the conclusions in this section seem overly confident for in silico analyses. Since GO and KEGG analysis are similarity-based rather than experimental, it is important to be cautious with the claims made. For example, the presence of domains found in transcription factors suggests potential activity, but it is not definitive without biochemical validation. Statements like "could also bind to small molecules, organic cyclic compounds, and proteins" or "enrichment in positively regulating by polymerase II" should be softened. The same caution applies to the subcellular localization predictions, as post-translational modifications can influence localization, which is not accounted for in standard prediction software. I recommend revising this section to ensure the limitations of these in silico methods are acknowledged.

15. Lines 273-278: Ensure that names are not italicized, and correct "Table x" to the appropriate reference.

16. Line 277: The word "beautified" is not ideal. Consider using terms like "plotted" or "edited for clarity."

17. Line 289: Correct "SPRING" to "STRING" in the legend.

18. Line 297: To help readers better understand the experiment, consider briefly describing the stressors, such as "(PEG)-induced drought (10% PEG), heat (40ºC), and light (continuous X μmol.s-1.m-²)."

19. Lines 331-341: It would strengthen the discussion to compare the data with other species. Are the protein sizes (in amino acids) similar across species? Do biochemical properties differ? Are there notable differences in subcellular localization or hydrophobicity? Addressing these points could provide biological insights or indicate limitations of the in silico methods.

20. Lines 227-228: Simply being present in the nucleus is insufficient to claim that PbCAMTAs influence transcription, let alone whether they induce or repress it. It would be better to soften this statement until experimental evidence supports the claim.

21. Line 396: Instead of saying the genes "were associated with light responsiveness," consider stating, "These genes have strong sequence-based evidence suggesting a potential response to light."

22. Lines 397-399: Similar to the previous point, be cautious when stating that these genes respond to various elicitors, as this conclusion is based on in silico evidence rather than experimental validation.

23. Lines 428-449: The manuscript does not clearly connect CAMTAs with the cited mechanisms of stress tolerance. How could CAMTAs play a role in these mechanisms? How could this be investigated in future studies? How might the findings of this study be applied to address challenges in P. bournei cultivation?

24. Figure 10: While speculative, the model provides a useful illustration of potential CAMTA functions. In the legend, indicate which stressors each square refers to, and consider highlighting the stressors (e.g., by using bold font) to make the diagram clearer. Also, mention this figure in the discussion, particularly in the context of the potential roles suggested in this section.

25. Lines 460-461: It is unclear whether the HMM models were obtained from Pfam or if Pfam was used for searches. Please clarify.

26. Line 461: Correct "Hummer" to "Hmmer."

27. Lines 462-463: How were the Arabidopsis CAMTAs selected? Were they based on a specific study or database search? Please clarify.

28. Line 464: The blastp searches do not confirm the identity of PbCAMTAs. It might be more accurate to state that the high similarity between AtCAMTAs and the PbCAMTAs identified in this study suggests the Hmmer searches were accurate.

29. Line 469: Italicize P. bournei.

30. Line 473: "MEME (meme-suite.org)" is sufficient. Please cite MEME following the guidelines provided at https://meme-suite.org/meme/doc/cite.html.

31. Line 474: Why was the maximum number of motifs set to 10? What was the e-value cutoff for the motifs?

32. Line 476: I do not understand why the CDD Search was used again. It was previously described in an earlier section.

33. Section 4.3: Please ensure that database citations and access dates are formatted correctly, such as "TAIR (arabidopsis.org, accessed July 1, 2024)" or "Ensembl Plants (plants.ensembl.org, accessed June 29, 2024)." Verify and format all citations consistently throughout the manuscript.

34. Section 4.3: How were the sequences chosen? Were they based on prior research or through software searches like BLAST? This should be clarified. Also, why was the Neighbor-Joining method chosen over Maximum Likelihood, and what were the specific values for G and F? What was the bootstrap cutoff for valid branches?

35. Lines 505-506: Was only R base used for visualization, or were specific packages, such as ggplot2, employed?

36. Line 518: Please avoid using the term "beautify." Consider "improve" or "refine" instead.

37. Line 522: Why did the control group contain 30 individuals while the treatment groups had only 3?

38. Section 4.7: Did the control group receive distilled water without nutrient solution? If so, this would not be a fair comparison. The only variable during the osmotic stress should be the presence of PEG.

39. PEG Molecular Weight: Please specify the molecular weight (MW) of the PEG used, as this affects the level of osmotic stress.

40. Stress Duration: Indicate how long the plants were exposed to each temperature condition.

41. Photoperiod: Was the photoperiod maintained during the stress treatments (except for the light treatment)? Please provide the photoperiod regime and light intensity, preferably in PPFD (μmol.s-1.m-²).

42. Leaf Sampling: Were all leaves collected from the same plants? If so, were they collected from the same leaf position (e.g., same leaf number from the apex)? Sampling from different leaf positions can lead to variability, as leaf physiology can differ based on position. For consistency, it would be ideal to sample the same leaf at each time point. Additionally, collecting samples from the control group at the same time points as the treatments would ensure consistency, since circadian rhythms also influence gene expression.

43. Section 4.8: Please correct the volumes listed for the qPCR reaction from liters (L) to microliters (µL). Additionally, correct "pre-degeneration" to "denaturation."

Comments on the Quality of English Language

The quality of the writing is high, but the manuscript would benefit from a more objective tone, cutting down excess words. It is also important to avoid  embellishments or overly decorative language, such as the use of "beautify" (lines 277 and 518), "was undertaken" (line 516), in a couple examples.

Author Response

Response

Comments and Suggestions for Authors

The presented manuscript had the goal to uncover and begin the characterization of CAMTA family genes in the ornamental and wood source species Phoebe bournei. The authors successfully identified 17 of these genes in the current assembly of the P. bournei genome, and have sufficient data to demonstrate their conservation in comparison to CAMTAs of other species. Nonetheless, some experimental details are unclear and some statements are overconfident, demanding more attention. Below are comments that will hopefully aid in improving the manuscript prior to publication.

  1. Line 66: There is a repetition of the word "Generally" in this line. Please revise for clarity.

Response1:We have deleted the word and the following sentence for a consideration of conciseness.

  1. Lines 79-80: The referenced study (reference 29) identified CAMTAs in five specific Solanaceae species: Solanum melongena, Solanum lycopersicum, Solanum pennellii, Capsicum annuum, and Lycium barbarum, with a focus on S. melongena. To avoid confusion, I suggest changing "in Solanaceae" to "in five different Solanaceae species" or specifying "Solanum melongena" to accurately reflect the study's content. This will help clarify that not all Solanaceae species were examined, such as Nicotiana tabacum, mentioned in line 76, which was not part of the study in reference 29.

Response2: We have specified "Solanum melongena" in “Solanum melongena, Solanum lycopersicum, Solanum pennellii, Capsicum annuum, and Lycium barbarum” Line 78-79.

  1. Lines 85-98: When discussing the functions of CAMTAs, it would be helpful to briefly mention the species along with the gene names. For instance, "GmCAMTA12 is an important transcription factor in soybean that can be exploited..." or "AtCAMTA6 in Arabidopsis thaliana modulates...". This would make it easier for readers unfamiliar with the gene abbreviations to follow the text.

Response3: Species have been stated when mentioning their genes.

  1. Line 113: The phrase "on Expasy" can be omitted, as it should be included in the methods section.

Response4: "on Expasy" have been omitted.

  1. Line 122: Consider revising "are considered instability" to "are considered unstable."

Response5: "are considered instability"have been revised to "are considered unstable." Line 133.

  1. Lines 130-132: It would be clearer to describe the sequence analysis software and method only in the methods section to avoid redundancy.

Response6:We have omitted software and method mentioned in lines 130-132

  1.  It was unclear to me which genes belong to each subfamily. Is this information provided in Figure 1?

Response7: The origin of "subfamily" was initially attributed to the intraspecific phylogenetic tree analysis presented in 2.2. However, after careful consideration, we have opted to discard this notion in order to avoid any potential confusion with the phylogenetic tree constructed between  P. bournei and other species in Section 2.4.

  1. Figure 1: The text in Figure 1 is difficult to read. Please consider increasing the font size for clarity. Additionally, indicate in the legend that the colored boxes in the phylogenetic tree represent MEME motifs in panel A (this is currently in panel B). Placing the color legend next to the tree in panel A would help avoid confusion with the colored boxes in panel D (representing UTR and CDS). The color legend in panel D is also incorrect—yellow represents UTRs and green represents CDS.

Response 8: Size of the characters have been amplified. Each legend is placed at the bottom right of its corresponding graph and the mistakes of color statements have been corrected.

  1. Lines 153-158 and Figure 2: The statements regarding the similarity of three-dimensional structures would be strengthened by performing structural alignment. I recommend using tools such as MatchMaker (doi:10.1186/1471-2105-7-339) from ChimeraX (doi:10.1002/pro.3943) and/or TM-align (doi:10.1093/nar/gki524) to calculate pairwise structural alignment scores. This would add more experimental confidence to the claims in this section.

Response9: We have used MatchMaker from ChimeraX and TM-align to predict the 3D structure of PbCAMTAs and found that its result is similar to SWISS-MODEL from Expasy, so we decided to continue using the prediction results from Swiss.The similarity of three-dimensional structures have been listed in a Table S2 which have been uploaded in supplementary data.

  1. Lines 173-175: I suggest revising the phrasing for clarity: "In groups I, II, and IV, there are only two PbCAMTAs each (PbCAMTA 6 and PbCAMTA 7, PbCAMTA 8 and PbCAMTA 9, PbCAMTA 4 and PbCAMTA 5, respectively)."

Response10: Sentence have been revised the sentence based on your suggestions. Line 172-174.

  1. Figure 4 (B and C): The font size and resolution need improvement, as the text is difficult to read.

Response11: The font size and resolution have been improved.

  1. Line 211: There may be a missing "and" before "monocotyledons."

Response12: We have corrected the sentence by adding "and" in front of "monocotyledons". Line 214.

  1. Lines 212-213: The species names should be italicized.

Response13:We have checked the entire text to ensure all our species were correctly italicized.

  1. Section 2.8: Some of the conclusions in this section seem overly confident for in silico analyses. Since GO and KEGG analysis are similarity-based rather than experimental, it is important to be cautious with the claims made. For example, the presence of domains found in transcription factors suggests potential activity, but it is not definitive without biochemical validation. Statements like "could also bind to small molecules, organic cyclic compounds, and proteins" or "enrichment in positively regulating by polymerase II" should be softened. The same caution applies to the subcellular localization predictions, as post-translational modifications can influence localization, which is not accounted for in standard prediction software. I recommend revising this section to ensure the limitations of these in silico methods are acknowledged.

Response14: Considering your comments, we have revised the inferences that were overconfident and not verified by definitive biochemical experiments. Meanwhile, in order to better understand subcellular localization, we have added the prediction results of transmembrane structures in Table S2, hoping it can improve our understanding of PbCAMTA. Section 2.8.

  1. Lines 273-278: Ensure that names are not italicized, and correct "Table x" to the appropriate reference.

Response15:We have specifically indicated the serial numbers of the tables in our text.

  1. Line 277: The word "beautified" is not ideal. Consider using terms like "plotted" or "edited for clarity."

Response16: All "beautified" in our manuscript have been according to your suggestion.

  1. Line 289: Correct "SPRING" to "STRING" in the legend.

Response17:We have corrected the legend.

  1. Line 297: To help readers better understand the experiment, consider briefly describing the stressors, such as "(PEG)-induced drought (10% PEG), heat (40ºC), and light (continuous X μmol.s-1.m-²)."

Response18: Section 4.7 have been rewritten. More details of the experiment have been correspondingly supplemented and described.

  1. Lines 331-341: It would strengthen the discussion to compare the data with other species. Are the protein sizes (in amino acids) similar across species? Do biochemical properties differ? Are there notable differences in subcellular localization or hydrophobicity? Addressing these points could provide biological insights or indicate limitations of the in silico methods.

Response19: We have compared the differences between CAMTA identified in our species and those identified in other species.

  1. Lines 227-228: Simply being present in the nucleus is insufficient to claim that PbCAMTAs influence transcription, let alone whether they induce or repress it. It would be better to soften this statement until experimental evidence supports the claim.

Response20: As you have suggested, the inference was indeed too hasty and we had revised the text to tone down the expression of this inference.

  1. Line 396: Instead of saying the genes "were associated with light responsiveness," consider stating, "These genes have strong sequence-based evidence suggesting a potential response to light."

Response21:We have corrected.L391-392

  1. Lines 397-399: Similar to the previous point, be cautious when stating that these genes respond to various elicitors, as this conclusion is based on in silico evidence rather than experimental validation.

Response22: The expression has been modified accordingly

  1. Lines 428-449: The manuscript does not clearly connect CAMTAs with the cited mechanisms of stress tolerance. How could CAMTAs play a role in these mechanisms? How could this be investigated in future studies? How might the findings of this study be applied to address challenges in P. bournei cultivation?

Response23: We are very sorry that this paragraph was not written well enough and it have been rephrased. As a transcription factor, PbCAMTA played a pivotal role in plants by specifically regulating target genes, thereby modulating plant stress responses and growth development. In our research, we have identified several genes (such as PbCAMTA8, PbCAMTA12) that respond to environmental stresses. For future studies, we can validate their specific functions through gene functional verification techniques, including gene knockout, overexpression, and other molecular biology methods. High-throughput sequencing technologies (ChIP-seq, RNA-seq and so on) can be employed to analyze the binding sites of this transcription factor on the genome and the target genes it regulates. Investigating the functions and mechanisms of transcription factors in plants can aid us in leveraging genetic engineering breeding techniques to introduce transcription factor genes with superior stress resistance into P. bournei, thereby cultivating new varieties with enhanced environmental adaptability. Alternatively, we can harness the mechanism of transcription factors regulating immune responses to develop novel biological pesticides. These products can activate the P. bournei innate immune system to combat pathogen infections, enhancing the disease resistance and stress tolerance of P. bournei.

  1. Figure 10: While speculative, the model provides a useful illustration of potential CAMTA functions. In the legend, indicate which stressors each square refers to, and consider highlighting the stressors (e.g., by using bold font) to make the diagram clearer. Also, mention this figure in the discussion, particularly in the context of the potential roles suggested in this section.

Response24: We have indicated in the captions which stressor each colored box refered to. The stressors in the figure have been emphasized by using bold font. We have also mentioned this figure in our discussion.

  1. Lines 460-461: It is unclear whether the HMM models were obtained from Pfam or if Pfam was used for searches. Please clarify.

Response25: HMM models were obtained from Pfam in order to obtain the sequences of the aiming domain. We have rephrase it to avoid further confusion in the text.

  1. Line 461: Correct "Hummer" to "Hmmer."

Response26:We have corrected.L456

  1. Lines 462-463: How were the Arabidopsis CAMTAs selected? Were they based on a specific study or database search? Please clarify.

Response27: They were selected from a database search.

  1. Line 464: The blastp searches do not confirm the identity of PbCAMTAs. It might be more accurate to state that the high similarity between AtCAMTAs and the PbCAMTAs identified in this study suggests the Hmmer searches were accurate.

Response28:We have corrected.L464-465

  1. Line 469: Italicize P. bournei.

Response29:We have corrected.Line 470

  1. Line 473: "MEME (meme-suite.org)" is sufficient. Please cite MEME following the guidelines provided at https://meme-suite.org/meme/doc/cite.html.

Response30:We have confirmed and corrected it in Line 470

  1. Line 474: Why was the maximum number of motifs set to 10? What was the e-value cutoff for the motifs?

Response31: We have listed motif sequences and their e-value in Table S1 in supplementary data. The e-values of these domains were all very low and the results were relatively reliable. However,  we refrained from searching for more motifs to ensure precision.

  1. Line 476: I do not understand why the CDD Search was used again. It was previously described in an earlier section.

Response32: We apologize for the redundancy in the information and have already removed the unnecessary details from the text. Additionally, we have revised and improved the majority of the expressions in Section 4.1.

  1. Section 4.3: Please ensure that database citations and access dates are formatted correctly, such as "TAIR (arabidopsis.org, accessed July 1, 2024)" or "Ensembl Plants (plants.ensembl.org, accessed June 29, 2024)." Verify and format all citations consistently throughout the manuscript.

Response33: All the dates mentioned in the text have been checked and revised according to your correct instructions. Section 4.3.

  1. Section 4.3: How were the sequences chosen? Were they based on prior research or through software searches like BLAST? This should be clarified. Also, why was the Neighbor-Joining method chosen over Maximum Likelihood, and what were the specific values for G and F? What was the bootstrap cutoff for valid branches?

Response34: The sequence were chose from prior researches. We have include this information to our manuscript. The phylogenetic tree was constructed with MEGA software using the Neighbor-Joining method and the Poisson correction model. Specific parameters were set as the Jones–Taylor–Thornton (JTT) model and 1000 bootstrap replications (with a bootstrap cutoff of 50),so there was no specific values for G and F. The reason why we choose Neighbor-Joining over Maximum Likelihood was that the Neighbor-Joining method is more suitable for relatively conserved sequences. For relatively conserved sequences, the results of the two methods do not differ much. There are previous studies also used the same method to analyze gene families. The articles are as follows:

Ahmad, F.; Rehman, S.U.; Rahman, M.H.U.; Ahmad, S.; Khan, Z. Characterization of Strubbelig-Receptor Family (SRF) Related to Drought and Heat Stress Tolerance in Upland Cotton (Gossypium hirsutum L.). Agronomy 2024, 14, 1933. https://doi.org/10.3390/agronomy14091933

AMA Style

Hussain, Q.; Ye, T.; Shang, C.; Li, S.; Nkoh, J.N.; Li, W.; Hu, Z. Genome-Wide Identification, Characterization, and Expression Analysis of the Copper-Containing Amine Oxidase Gene Family in Mangrove Kandelia obovata. Int. J. Mol. Sci. 2023, 24, 17312.

Aizaz, M.; Kiani, Y.S.; Nisar, M.; Shan, S.; Paracha, R.Z.; Yang, G. Genomic Analysis, Evolution and Characterization of E3 Ubiquitin Protein Ligase (TRIM) Gene Family in Common Carp (Cyprinus carpio). Genes 2023, 14, 667.

  1. Lines 505-506: Was only R base used for visualization, or were specific packages, such as ggplot2, employed?

Response35: R was also used for statistic analysis, ggplots was indeed used. Both have been stated according to the text.

  1. Line 518: Please avoid using the term "beautify." Consider "improve" or "refine" instead.

Response36: We have corrected.L514

  1. Line 522: Why did the control group contain 30 individuals while the treatment groups had only 3?

Response37: We are sorry for our careless experimental instructions. In our experiment, we sampled different seedlings at 5 time points for drought and heat treatments respectively, while for light treatment, we sampled different seedlings at 4 time points (including three biological replicates for each), so there were 42 individuals in total. We used the 0h of the treatment group as the control and made comparisons. Such experimental setups have also been used in the following articles.

Li Z, Yang Y, Chen B, Xia B, Li H, Zhou Y, He M. Genome-wide identification and expression analysis of SBP-box gene family reveal their involvement in hormone response and abiotic stresses in Chrysanthemum nankingense. PeerJ. 2022 Oct 27;10:e14241. doi: 10.7717/peerj.14241. PMID: 36320567; PMCID: PMC9618261.

Song, N.; Cheng, Y.; Peng, W.; Peng, E.; Zhao, Z.; Liu, T.; Yi, T.; Dai, L.; Wang, B.; Hong, Y. Genome-Wide Characterization and Expression Analysis of the SBP-Box Gene Family in Sweet Orange (Citrus sinensis). Int. J. Mol. Sci. 2021, 22, 8918. https://doi.org/10.3390/ijms22168918

Ghorbel, M.; Zribi, I.; Besbes, M.; Bouali, N.; Brini, F. Catalase Gene Family in Durum Wheat: Genome-Wide Analysis and Expression Profiling in Response to Multiple Abiotic Stress Conditions. Plants 2023, 12, 2720.

  1. Section 4.7: Did the control group receive distilled water without nutrient solution? If so, this would not be a fair comparison. The only variable during the osmotic stress should be the presence of PEG.

Response38: As responded in Comment 37, seedling treated for 0 hour which could be regarded as the untreated group, as well as a control group. As a untreated group, they were not distilled into any solution.

  1. PEG Molecular Weight: Please specify the molecular weight (MW) of the PEG used, as this affects the level of osmotic stress.

Response39: Thank you for your reminder. We have indicated that the treatment group was transplanted into a beaker containing 10% PEG 6000 of ¼-strength Hogland solution

  1. Stress Duration: Indicate how long the plants were exposed to each temperature condition.

Response40: We have only conducted high temperature treatments and the experiment description of 10 degrees was an mistake. We have corrected and rephrased this in section4.7. The heat treated group were sampled at 0, 4, 8, 12, and 24 hours while the individuals sampled at 0 hour were regarded as the control group. Section 4.7.

  1. Photoperiod: Was the photoperiod maintained during the stress treatments (except for the light treatment)? Please provide the photoperiod regime and light intensity, preferably in PPFD (μmol.s-1.m-²).

Response41: Indeed, photoperiod (12h/d) maintained during the stress treatments. More detailed information about our experiment have been provided in section 4.7.

  1. Leaf Sampling: Were all leaves collected from the same plants? If so, were they collected from the same leaf position (e.g., same leaf number from the apex)? Sampling from different leaf positions can lead to variability, as leaf physiology can differ based on position. For consistency, it would be ideal to sample the same leaf at each time point. Additionally, collecting samples from the control group at the same time points as the treatments would ensure consistency, since circadian rhythms also influence gene expression.

Response42: Each different treatment and sampling time point corresponds to a group of plants, and each group includes three biological replicates. We collect samples from each individual plant, specifically targeting leaves located at the central position. We regarded the 0h from the treatment group as the control and made comparisons.. Seedlings cultivated for one year are very precious therefore we did not set up an independent control group. We relied on sampling the untreated control group (0h) with 3 technical replicates to ensure the reliability of the results. In addition, other studies have also adopted such research schemes.

  1. Section 4.8: Please correct the volumes listed for the qPCR reaction from liters (L) to microliters (µL). Additionally, correct "pre-degeneration" to "denaturation."

 Response43: Both mistakes have been corrected. Line 559-560.

Comments on the Quality of English Language

The quality of the writing is high, but the manuscript would benefit from a more objective tone, cutting down excess words. It is also important to avoid  embellishments or overly decorative language, such as the use of "beautify" (lines 277 and 518), "was undertaken" (line 516), in a couple examples

Response: We have reviewed our manuscript and rewritten the overconfident conclusions as more objective speculations. All the "beautified" in our manuscript have been modified based on your suggestions.

Round 2

Reviewer 1 Report

Comments and Suggestions for Authors

Most of my previous objections have been answered, but there are still some things to be improved - I have highlighted them in attached file.

And one of the explanation from cover letter - "Response15: corrosion resistance presented as ability to resistant extreme acid base environment, insects and etc. We have made a specific explanation of this word in the text so that readers can understand it better. Line 319." - does not exist in the text.

Comments on the Quality of English Language

As I have highlighted minor editing of English language is required.

Reviewer 2 Report

Comments and Suggestions for Authors

The authors did a good job in revising the MS. However, some more changes are required before it can be accepted for publication. My comments are summarized below:

Minor comments:

1. Line 109-110: Add gene before expression as it is gene expression profile and remove heat as you have heat and heat stress.

2. Line 131-132: Subcellular localization is for protein and not gene so don’t italicize as proteins are regular and italicize if the mentioned is a gene.

3. Line 145-150: Authors need to define all the abbreviations in the first place as they appear. Check throughout the MS.

4. Line 235: 2.0 kp promoter. It should be kbp and italicize PbCAMTA as you are saying it is a gene.

5. Line 272: Figure 6 caption, PbCAMTA

6. Table 2: Italicize the gene names and the same in Table S5.

7. Line 551: Cite the paper for 2DDCT and fold difference calculations. See the below article for these references.

Rathor, P., Borza, T., Stone, S., Tonon, T., Yurgel, S., Potin, P., & Prithiviraj, B. (2021). A novel protein from Ectocarpus sp. improves salinity and high temperature stress tolerance in Arabidopsis thaliana. International Journal of Molecular Sciences, 22(4), 1971. https://doi.org/10.3390/ijms22041971

Pfaffl MW (2001) A new mathematical model for relative quantification in real-time RT-PCR. Nucleic Acids Res 29:e45 https://doi.org/10.1093/nar/29.9.e45

8. References: check as some journal titles are all uppercase and many are lowercase. For example, No. 7, 13, 16, 30 and more. Some journals are abbreviated, and some are not. Be consistent for all the references.

Comments on the Quality of English Language

English editing has been done but still some grammatical errors and typos error are present.

Reviewer 3 Report

Comments and Suggestions for Authors

The revised manuscript addresses all the points raised in the review and demonstrates significant improvement.

Author Response

Thank you very much for your previous suggestions and guidances.